# Rewiring Experts on the Fly: Continuous Rerouting for Better Online Adaptation in Mixture-of-Expert models

## Abstract

Mixture-of-Experts (MoE) models achieve efficient scaling through sparse expert activation, but often suffer from suboptimal routing decisions due to distribution shifts in deployment. While existing test-time adaptation methods could potentially address these issues, they primarily focus on dense models and require access to external data, limiting their practical applicability to MoE architectures. However, we find that, instead of relying on reference data, we can optimize MoE expert selection on-the-fly based only on input context. As such, we propose *a data-free, online test-time framework* that continuously adapts MoE routing decisions during text generation without external supervision or data. Our method cycles between two phases: During the prefill stage, and later in regular intervals, we optimize the routing decisions of the model using self-supervision based on the already generated sequence. Then, we generate text as normal, maintaining the modified router until the next adaption. We implement this through lightweight additive vectors that only update router logits in selected layers, maintaining computational efficiency while preventing over-adaptation. The experimental results show consistent performance gains on challenging reasoning tasks while maintaining robustness to context shifts. For example, our method achieves a 5.5% improvement on HumanEval with OLMoE. Furthermore, owing to its plug-and-play property, our method naturally complements existing test-time scaling techniques, e.g., achieving 6% average gains when incorporated with self-consistency on DeepSeek-V2-Lite.

## 1 Introduction

Mixture-of-Experts (MoE) models (Shazeer et al., 2017; Zhou et al., 2022; Jiang et al., 2024; Dai et al., 2024; Liu et al., 2024; Team, 2025; Muennighoff et al., 2024) provide an effective approach to scaling model capacity while maintaining computational efficiency by partitioning parameters into specialized experts and selectively activating subsets through routing mechanisms (Lepikhin et al., 2020; Fedus et al., 2022; Dai et al., 2024; Muennighoff et al., 2024). This functionality enables dynamic expert selection for diverse queries and creating inherently general-purpose systems that can store much more functionality and information than is used in every forward pass. However, despite their impressive capabilities, MoE models still face challenges when deployed in real-world environments (Akyürek et al., 2024; Li et al., 2025a), as the expression of their capability hinges on the quality of their *routing decisions*, the activations of small linear layers that determine which parts of the model are activated.

Why is routing hard? While the full MoE may store sufficient functionality to solve a particularly challenging query, this capacity is gated behind its routing decisions in each MoE layer, which in turn depend on the residual stream of the model, and so on earlier routing decisions. Routing decisions are only linear functions of the current hidden state that need to approximate the anticipated utility of activating a certain expert. A particular issue with this non-robustness of routing is that during standard inference *there is no mechanism to reinforce the routing to a particularly successful part of the model*, or to reduce routing to parts that did not contribute meaningful signal to the generated text.

This makes routing optimization a question of model adaptation, reminiscent of neuroplasticity in humans – who continuously optimize routing and neuronal connections in the brain through adaptation and self-regulation, for example when continuing to practice a certain task. Yet, in MoEs,

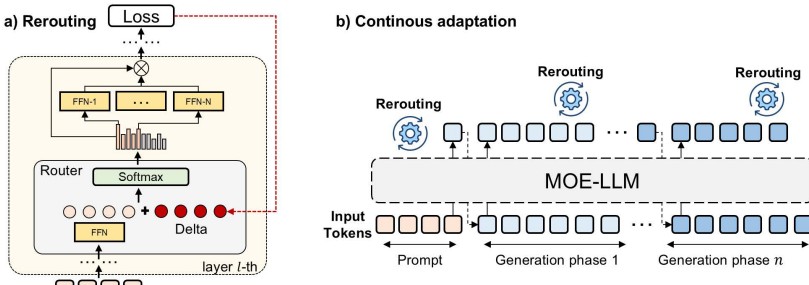

**Figure 1: Test-time rerouting framework for MoE models.** (a) **Rerouting mechanism**: lightweight additive vectors (Delta) update router logits in selected high-confidence layers using self-supervised loss from existing context. (b) **Continuous adaptation**: alternating between optimization phases that adapt routing decisions and generation phases that maintain adapted routing until the next optimization cycle.

suboptimal expert selection leads to routing inefficiencies, creating a critical bottleneck in overall performance (Shi et al., 2024; Li et al., 2025a). And, while there has been a significant amount of work to improve the capabilities of MoEs in general, such as through test-time scaling, it is less clear how to best adapt MoEs to different tasks at inference. While standard approaches, such as in-context learning with task-specific demonstrations (Wei et al., 2022; Madaan et al., 2023) or parallel generation strategies that produce multiple candidates and aggregate them (Wang et al., 2022; Brown et al., 2024), do adapt the model overall, they do influence routing only implicitly.

Conceptually, adaptation should be solvable by *test-time training*, but existing approaches (Hardt & Sun; Hübotter et al., 2024) focus on a classical prediction perspective, which retrieves relevant data from training sets or knowledge bases during inference to fine-tune models for dynamic scenarios before the model is being used. Li et al. (2025a) attempt to address router optimization through this perspective of test-time training, and, as such, use retrieval of "successful neighbors" from reference sets based on the first prompt in each context. However, this approach requires access to external reference data during deployment, incurs retrieval overhead, and risks failures in retrieval due to short prompts. Moreover the approach is static, as modern models execute test-time scaling, they generate long chains-of-thought, that should be taken into account when optimizing routing.

In this regard, we propose a simple yet effective *data-free* test-time rerouting framework for MoE models as shown in Figure 1 that treats each input prompt as a self-supervised learning opportunity, enabling dynamic rerouting of pretrained MoE models for individual prompts during inference. The framework alternates in two phases: (1) **In-Context Routing Optimization**, where we regard the current context itself as a training sample and execute optimization steps to minimize cross-entropy loss on the current context with regard to routing logits; and (2) **Steered Generation**, where we generate text normally, steering routers with updates computed in the previous phase. This creates a dynamic feedback loop where *the model continuously refines its understanding of task requirements* based on its own generation progress, enabling increasingly informed expert routing decisions as generation proceeds. To maintain computational efficiency, we implement this progressive optimization through **lightweight additive parameter vectors that update only the model's router logits of selected MoE layers**, further reducing computational overhead and preventing over-adaptation. Overall, our main contributions are:

- We propose a test-time rerouting framework specifically designed for MoE models, operating completely without external data dependencies or expensive retrieval mechanisms, using only backpropagation within the current context.

- We introduce a lightweight parameter update mechanism that selectively optimizes router logits through additive vectors, enabling efficient expert selection adaptation during inference using steering.

- We validate our method's effectiveness through extensive experiments, demonstrating that our approach significantly improves MoE model performance on complex reasoning tasks while maintaining computational efficiency and generation diversity.

Overall, we argue that routing changes are a compelling mechanism with which to adapt MoE models on the fly. The changes are lightweight, quick to compute and to apply (even in multi-user

settings) and allow MoEs a novel degree of plasticity that allows adaptation to task changes during deployment, paving the way toward practical continual self-regulation of MoE models during use.

## 2 RELATED WORK

**Mixture-Of-Experts (MoE).** Sparsely activated Mixture-of-Expert models (MoE) are an efficient method for scaling model size by replacing the feed-forward networks (FFN) of transformer architectures with multiple experts (each its own FFN) and a gating function. This architecture dynamically activates different experts for each input token rather than utilizing all parameters (Shazeer et al., 2017; Lepikhin et al., 2020; Fedus et al., 2022). Recent MoE-based LLMs, including OL-MoE (Muennighoff et al., 2024), DeepSeek (Dai et al., 2024; Liu et al., 2024), and Qwen2.5 (Team, 2025), adopt top-k expert routing to reduce the active parameter count during inference. These models achieve competitive performance while maintaining significantly lower memory costs.

**Test-Time Scaling.** Test-Time Scaling enhances LLM capabilities by allocating additional computational resources during inference. These approaches fall into two categories (Welleck et al., 2024): Parallel generation, including self-consistency evaluations via multiple candidate responses (Wang et al., 2022), best-of-N sampling (Brown et al., 2024), and Monte Carlo Tree Search (Zhou et al., 2023; Xie et al., 2024), and sequential generation such as extending outputs through chain-of-thought reasoning (Wei et al., 2022; Madaan et al., 2023).

**Test-Time Training (TTT).** TTT offers an alternative scaling approach. While successful in computer vision (Wang et al., 2020; Sun et al., 2020; 2024; Gandelsman et al., 2022; Osowiechi et al., 2023), recent works have extended TTT to language models through fine-tuning on retrieved neighbors (Hardt & Sun) or optimized data selection algorithms like SIFT (Hübotter et al., 2024). Test-Time Reinforcement Learning (TTRL) (Zuo et al., 2025) uses majority voting as reward signals. When applied to Mixture-of-Experts (MoE) models, recent work has explored expert-level interventions for behavior control, enabling precise modifications through targeted expert manipulation (Dahlke et al., 2025; Wang et al., 2025). Most relevant to our approach, Li et al. (2025a) optimizes expert routing in MoE models using "successful neighbors" from reference sets. However, these methods assume accessible training data during deployment and introduce significant retrieval overhead, limiting real-world practicality.

## 3 METHODOLOGY

This section presents our data-free test-time rerouting framework for MoE models that optimizes expert routing during inference without external information. After reviewing MoE fundamentals (Section 3.1), we introduce three key components: (1) Router Logits Modification (Section 3.2) for layer-specific expert selection steering, (2) Dynamic Layer Selection (Section 3.3) for selective MoE layer updates based on confidence scores, and (3) Optimization Procedure (Section 3.4) detailing our two-phase strategy alternating between in-context routing optimization and steered generation.

### 3.1 PRELIMINARIES ON MIXTURE-OF-EXPERTS

In Transformer-based Mixture-of-Experts (MoE) models, the conventional Feed-Forward Networks (FFNs) are replaced with MoE layers. Each MoE layer consists of a router $R$ and a set of experts $\{E_i\}_{i=1}^N$. Given an input sequence $\mathbf{x}_{<t} = (x_1, x_2, \ldots, x_{t-1})$, the router assigns each token to a subset of experts for processing. Given a token's hidden state $\mathbf{h} \in \mathbb{R}^d$ at layer $l$, the router computes logits across all $N$ experts: $\mathbf{z}^{(l)} = W_r^{(l)}\mathbf{h}^{(l)}$ where $W_r^{(l)} \in \mathbb{R}^{N \times d}$ is the router's weight matrix at layer $l$, and $\mathbf{z}^{(l)} \in \mathbb{R}^N$ represents the logits for expert selection. These logits are then converted to expert selection probabilities: $\mathbf{w}^{(l)} = \mathrm{Softmax}(\mathbf{z}^{(l)})$ where $w_i^{(l)}$ represents the activation probability for expert $E_i$ at layer $l$. The router applies a routing strategy (e.g., top-$k$ routing) to select active experts. Weights for unselected experts are zeroed and the remaining weights are renormalized to $\hat{\mathbf{w}}^{(l)}$. The final MoE output is: $\mathbf{o}^{(l)} = \sum_{i \in \mathcal{A}^{(l)}} \hat{w}_i^{(l)} \cdot E_i(\mathbf{h}^{(l)})$ where $\mathcal{A}^{(l)} = \{j \mid \hat{w}_j^{(l)} \neq 0\}$ denotes the set of activated experts at layer $l$.

## 3.2 ROUTER LOGITS MODIFICATION

We introduce layer-specific adaptation parameters $\{\boldsymbol{\delta}^{(l)}\}_{l=1}^{L}$ where $\boldsymbol{\delta}^{(l)} \in \mathbb{R}^N$ corresponds to the $N$ experts at MoE layer $l$. For a selected layer $l$, we modify the router logits by adding the corresponding layer-specific parameter: $\tilde{\mathbf{z}}^{(l)} = \mathbf{z}^{(l)} + \boldsymbol{\delta}^{(l)}$ where $\tilde{\mathbf{z}}^{(l)} \in \mathbb{R}^N$ represents the modified logits for layer $l$. The expert selection probabilities are then computed as:

$$\tilde{\mathbf{w}}^{(l)} = \text{Softmax}(\tilde{\mathbf{z}}^{(l)}) = \text{Softmax}(\mathbf{z}^{(l)} + \boldsymbol{\delta}^{(l)}) \tag{1}$$

This modification directly influences the expert selection distribution, allowing the model to adapt routing decisions based on prompt characteristics.

## 3.3 DYNAMIC LAYER SELECTION

To mitigate computational overhead and prevent over-adaptation, our framework selectively updates router logits of only a subset of MoE layers rather than all layers simultaneously. We hypothesize that layers with higher routing confidence indicate more decisive and task-relevant expert selection, making them more impactful for adaptation. We define router confidence at layer $i$ for token $n$ as:

$$C_i^{(n)} = -\frac{1}{k} \sum_{j=1}^{k} \log p_{i,j}^{(n)} \tag{2}$$

where $p_{i,j}^{(n)}$ is the probability of the $j$-th top expert at layer $i$ for token $n$, and $k$ is the number of activated experts per layer. Higher confidence values indicate more decisive routing decisions, suggesting these layers play more critical roles for the current task. To obtain layer-level confidence across the generated sequence, we aggregate token-level confidence:

$$C_i = \frac{1}{N} \sum_{n=1}^{N} C_i^{(n)} \tag{3}$$

where $N$ is the number of generated tokens so far. We implement two layer selection strategies: (1) **Hard selection** that selects top-$r$ proportion of layers with highest confidence scores: $\mathcal{S}_t = \text{TopK}(\{C_i\}_{i=1}^{L}, r)$, and (2) **Soft weighting** that assigns confidence-based weights to control the update strength of $\boldsymbol{\delta}^{(l)}$:

$$w_i = \frac{C_i}{\sum_{j=1}^{L} C_j} \tag{4}$$

During gradient updates, $\boldsymbol{\delta}^{(l)}$ is updated with learning rate scaled by $w_l$.

## 3.4 OPTIMIZATION PROCEDURE

**Parameter Initialization:** For each MoE layer $l \in \mathcal{L}$, initialize routing adjustment parameters:

$$\boldsymbol{\delta}^{(l)} = \mathbf{0} \in \mathbb{R}^N \tag{5}$$

The framework alternates between two phases:

**Phase 1: In-Context Routing Optimization.** Given current context $\mathbf{x} = (x_1, \ldots, x_t)$, first select layers using $\mathcal{S} = \text{TopK}(\{C^{(l)}\}_{l=1}^{L}, r)$ or compute soft weights $\{w_l\}_{l=1}^{L}$. Then perform $n$ optimization steps, where at each step we compute the cross-entropy loss:

$$\mathcal{L}(\{\boldsymbol{\delta}^{(l)}\}_{l=1}^{L}) = -\sum_{i=1}^{t-1} \log p(x_{i+1} \mid x_{1:i}, \{\boldsymbol{\delta}^{(l)}\}_{l=1}^{L}) \tag{6}$$

and update parameters using optimizer $\mathcal{O}$ (e.g., SGD, Adam):

$$\boldsymbol{\delta}^{(l)} \leftarrow \begin{cases} \mathcal{O}(\boldsymbol{\delta}^{(l)}, \nabla_{\boldsymbol{\delta}^{(l)}} \mathcal{L}) & \text{if } l \in \mathcal{S} \text{ (hard selection)} \\ \mathcal{O}(\boldsymbol{\delta}^{(l)}, w_l \nabla_{\boldsymbol{\delta}^{(l)}} \mathcal{L}) & \text{if soft weighting} \\ \boldsymbol{\delta}^{(l)} & \text{otherwise} \end{cases} \tag{7}$$

**Phase 2: Steered Generation.** Generate $m$ tokens using optimized routing parameters $\{\boldsymbol{\delta}^{(l)}\}_{l=1}^{L}$.

After generating m tokens, return to Phase 1 with extended context including both original prompt and all generated tokens up to the current position and repeat the optimization procedure. The detailed algorithm is provided in Algorithm 1.

## 4 EXPERIMENTS

**Benchmarks** We evaluate our approach using a diverse set of benchmarks across multiple reasoning domains. For general knowledge assessment, we employ MMLU-redux(Gema et al., 2024), utilizing generation mode rather than multiple-choice format to encourage deeper reasoning before providing final answers. For code generation tasks, we use HumanEval (Chen et al., 2021) and MBPP-sanitized (Austin et al., 2021). For mathematical reasoning, we evaluate on GSM8K (Cobbe et al., 2021) and MATH500 (Lightman et al., 2023).

**Baselines** We compare our method with two adaptation techniques: In-Context Learning (ICL) (Wei et al., 2022; Madaan et al., 2023) and C3PO (Li et al., 2025a). Since our method is a data-free sequential generation test-scaling approach, we select In-Context Learning (ICL) as a primary baseline, using 3 and 5 sample pairs respectively, for comparison. We also compare against C3PO, for which we select a reference set of 100 samples for each dataset. Note that parallel generation methods represent an orthogonal approach to ours, and we further discuss the potential integration of our method with such approaches in later sections.

**Model Selection** We evaluate three MoE LLMs: OLMoE (Muennighoff et al., 2024), Qwen1.5-MoE (Team, 2024), and DeepSeek-V2-Lite (Liu et al., 2024). OLMoE employs 16 layers with 64 experts per layer, activating 8 experts per token (6.9B total, 1.3B active parameters). Qwen1.5-MoE-A2.7B, incorporates 4 shared experts alongside 60 routing experts with 4 activated per token (14.3B total, 2.7B active parameters). DeepSeek-V2-Lite uses 28 layers with 2 shared and 64 routed experts, activating all shared plus 6 routed experts per token (16B total, 2.4B active parameters).

**Optimization** We optimize the adaptation parameters using the Adam optimizer with a small number of iterations ($T = 5$) to maintain computational efficiency. The optimization hyperparameters are set as follows: learning rate $\eta = 0.05$, weight decay $= 1 \times 10^{-8}$, and epsilon $= 1 \times 10^{-5}$. All adaptation parameters are initialized to zero: $\boldsymbol{\delta}^{(l)} = \mathbf{0}$. We use the soft-weighting strategy for layer selection, and during the generation stage, we periodically re-optimize the parameters during generation using identical hyperparameter settings to ensure continuous refinement throughout the sequence generation process. Specifically, we set the re-optimization interval to 128 tokens for math and MMLU tasks, and 96 tokens for code generation tasks.

## 5 RESULTS

### 5.1 MAIN RESULTS

We first evaluated the performance of our method on challenging reasoning benchmarks. As shown in Table 1, our approach consistently improves performance over all baselines across five benchmarks. On the HumanEval task in particular, our method yields gains of 3.6%, 5.5%, and 6.7% on DeepSeek-V2-Lite, OLMoE, and Qwen1.5-MoE, respectively. Moreover, compared to other calibration methods such as few-shot learning and C3PO, our method achieves consistently better results. Notably, it surpasses these baselines without relying on example demonstrations, delivering particularly strong improvements in code generation and mathematical reasoning tasks. Moreover, in contrast to C3PO, which requires 100 reference samples, our data-free method attains superior performance while requiring no additional reference data.

### 5.2 ABLATION

To validate the effectiveness of our design choices, we conduct ablation studies on using DeepSeek-V2-Lite, examining two key components: layer selection strategies and continuous refinement mechanisms. We evaluate performance across five tasks, with results presented in Table 2.

**Table 1:** Model Performance Comparison Across Different Benchmarks

| Method | HumanEval | MBPP | GSM8K | MATH500 | MMLU | Average |
|---|---|---|---|---|---|---|
| **DeepSeek-V2-Lite** | | | | | | |
| Baseline | 50.60 | 58.37 | 72.10 | 22.60 | 50.77 | 50.89 |
| ICL (3-shot) | 52.44 | 56.81 | 71.10 | 21.60 | 44.33 | 49.26 |
| ICL (5-shot) | 53.05 | 52.53 | 71.57 | 22.20 | 46.07 | 49.08 |
| C3PO (100-reference) | 47.80 | 59.92 | 68.80 | 18.20 | 45.77 | 48.10 |
| **Ours** (0-shot) | **54.26** | **62.65** | **73.62** | **25.00** | **52.40** | **53.59** |
| **OLMoE** | | | | | | |
| Baseline | 28.66 | 40.08 | 51.48 | 10.40 | 37.17 | 33.56 |
| ICL (3-shot) | 28.05 | 36.96 | 43.82 | 9.90 | **44.90** | 32.73 |
| ICL (5-shot) | 21.21 | 39.30 | 44.96 | 9.40 | 43.63 | 31.70 |
| C3PO (100-reference) | 28.05 | 41.25 | 52.08 | **13.00** | 37.30 | 34.33 |
| **Ours** (0-shot) | **34.17** | **42.80** | **52.99** | 12.40 | 38.30 | **36.13** |
| **Qwen1.5-MoE** | | | | | | |
| Baseline | 40.24 | 46.69 | 54.73 | 21.80 | 45.27 | 41.75 |
| ICL (3-shot) | 44.34 | 46.25 | 55.27 | **22.20** | 45.60 | 42.73 |
| ICL (5-shot) | 43.90 | 44.75 | 52.31 | 19.40 | **46.03** | 41.28 |
| C3PO (100-reference) | 39.70 | 45.40 | 50.33 | 18.90 | 44.82 | 39.83 |
| **Ours** (0-shot) | **46.95** | **47.47** | **56.00** | 22.00 | 45.87 | **43.66** |

**Table 2:** Ablation Study: Layer Selection Strategies and Continuous Refinement

| Method | HumanEval | MBPP | GSM8K | MATH500 | MMLU | Average |
|---|---|---|---|---|---|---|
| Baseline | 50.60 | 58.37 | 72.10 | 22.60 | 50.77 | 50.89 |
| *Layer Selection Strategies* | | | | | | |
| Random Selection | 48.78 | 53.31 | 72.50 | 23.40 | 50.87 | 49.77 |
| Reverse Metric | 48.18 | 55.26 | 72.10 | 21.89 | 49.33 | 49.35 |
| Last-five Layers | 50.60 | 55.64 | 71.85 | 25.20 | 50.33 | 50.72 |
| All Layers | 48.17 | 56.42 | 73.09 | 24.60 | 51.10 | 50.68 |
| *Continuous Refinement* | | | | | | |
| W/o Continuous | 54.27 | 59.53 | 73.37 | 23.80 | 51.80 | 52.55 |
| **Ours** | **54.27** | **62.65** | **73.62** | **25.00** | **52.40** | **53.59** |

**Confidence-based layer selection efficiently discovers task-specific layers.** We compare our confidence-based selection with several baselines, including random selection, selecting the last five layers, selecting low-confidence layers (Reverse Metric), and optimizing all layers. As shown in Table 2, our method achieves an average performance of 53.59%, outperforming random selection (49.77%), the last-five-layer strategy used in C3PO (50.72%), and reverse selection targeting low-confidence layers (49.35%). This demonstrates that router confidence identifies layers with strong expert selection knowledge. In effect, we use the signal from prior context to "confirm" that routing choices were accurate, and that the model should rely on these experts more strongly.

**Updating Single Layers is Safer than rerouting the whole model.** In addition, from the results we can see that updating all layers simultaneously (50.68% on average) underperforms our selective approach (53.59% on average), highlighting the superiority of our method in selecting layers for test-time rerouting. With limited optimization steps available, spreading updates across all parameters dilutes the refinement signal and risks overadaptation that destabilizes pre-trained routing patterns. Concentrating the optimization budget on high-confidence layers maximizes impact by amplifying existing routing strengths rather than attempting comprehensive corrections across the entire network.

**Continuous Refinement Stabilizes Routing.** The comparison between continuous (53.59% on average) and non-continuous refinement (52.55% on average) reveals a meaningful performance gap of over 1%. It is notable that this gap carries weight, given that each benchmark example involves a single task and prompt-based router updates already constitute a strong baseline. This improvement indicates that routing optimization benefits from curriculum-like approaches that gradually refine the model's expert selection mechanisms, leading to more stable and effective adaptations during test-time optimization.

# 6 ANALYSIS AND DISCUSSIONS

## 6.1 WHY DOES TEST-TIME REROUTING WORK?

To understand the mechanisms behind test-time rerouting, we perform a comprehensive analysis of the DeepSeek-V2-Lite model, examining pathway shifts, expert utilization dynamics, and the evolution of routing confidence.

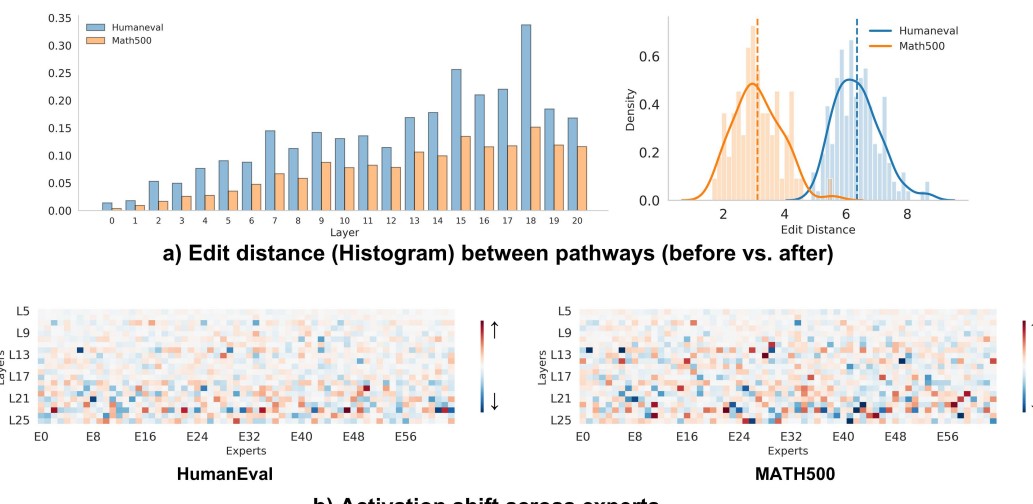

**a) Edit distance (Histogram) between pathways (before vs. after)**

**b) Activation shift across experts**

**Figure 2:** Analysis of test-time rerouting mechanisms across different datasets in DeepSeek-V2-Lite. a) Edit distance before and after rerouting. b) Expert utilization dynamics before and after rerouting.

**Rewiring improves Expert Pathways.** We first examine how expert pathways change after rerouting using edit distance following (Li et al., 2025b) (described in Appendix C.2), which quantifies pathway differences using Levenshtein edit distance. This captures mismatches in expert selection, and pathway shifts. We track the average pairwise edit distance across samples during rerouting.

The results are displayed in Figure 2 a), which shows distinct layer-wise editing patterns. In HumanEval, edit distances peak in deeper layers (18–22, >0.35) while earlier ones (5–10) remain minimal (<0.05). MATH500 exhibits a similar but stronger trend, with peaks up to 0.16 in layers 20–22, suggesting that deeper layers are more involved in mathematical reasoning. The sample-wise distributions (right panels) further emphasize task-specific differences, indicating high variability in pathway changes across problems. Overall, these results highlight the heterogeneity of expert routing across tasks and demonstrate how our method adaptively reroute pathways.

**Rewiring highlights Task-Specific Experts.** We further examine expert-level activation shifts on HumanEval and MATH500 before and after rerouting to understand how the model redistributes computation. Figure 2 b) shows heatmaps across layers and experts (red: increased, blue: decreased, normalized for visualization). The results reveal: (1) Adaptive targeting. changes concentrate on a subset of experts, indicating strategic rather than uniform redistribution; (2) Deep-layer adaptation. later layers (L21–L25) undergo stronger shifts, consistent with our edit-distance findings; and (3) Task-specific specialization. HumanEval and MATH500 exhibit distinct patterns, highlighting task-dependent routing behaviors. This selective activation shows that rerouting strategically emphasizes task-relevant experts, enhancing efficiency by focusing computation on where it is most needed.

**Rewiring Increases Routing Confidence.** Moreover, we quantify routing confidence by tracking the entropy of expert-selection distributions across all layers during optimization. Lower entropy reflects more focused expert allocation, whereas higher entropy indicates more diffuse routing patterns.

Figure 3 shows that our method (red line) exhibits a gradual decrease in entropy over 18 generation steps, whereas the baseline (blue line) maintains higher entropy with pronounced fluctuations. This suggests that our approach progressively develops more focused routing decisions, while the baseline continues to rely on more diffuse expert-selection patterns. This indicates that our method facil-

itates concentrating on relevant experts without disrupting established routing mechanisms, thereby enabling more efficient expert utilization for the given tasks.

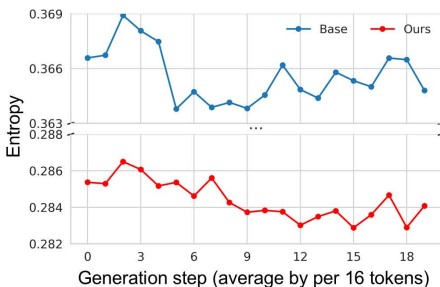

**Figure 3:** Expert routing entropy evolution during generation.

**Table 3:** Computational efficiency comparison across different methods.

| Method | Total FLOPs | Time (s) |
|---|---|---|
| Baseline | 4.71e+11 | 10.71s |
| ICL(3-shot) | 1.93e+12 | 12.17s |
| ICL(5-shot) | 3.05e+12 | 12.53s |
| Self-Consistency (3) | 1.68e+12 | 34.20s |
| C3PO(100-reference) | 2.63e+12 | 26.20s |
| Ours | 1.96e+12 | 20.12s |

## 6.2 COMBINING REWIRING WITH OTHER TEST-TIME STRATEGIES

A key advantage of our approach is its plug-and-play compatibility with existing test-time techniques. Since our method only modifies routing decisions without altering the underlying generation process, it can be seamlessly integrated with other approaches like in-context learning and parallel generation methods.

**Synergy with In-Context Learning.** Large pretrained language models demonstrate strong in-context learning, predicting labels from few demonstration pairs without parameter updates. Empirical evidence demonstrates behavior similar to explicit finetuning (Dai et al., 2022; Akyürek et al., 2022; Von Oswald et al., 2023). As such, we test combining our method with with 3-shot demonstrations. As shown in Table 4, we observe that this combination yields improvements across multiple tasks, even surpassing our method alone. We hypothesize this synergy occurs because our method provides more effective gradient updates that better leverage the contextual information from demonstration examples, helping extract more meaningful patterns from the limited demonstration data.

**Unlocking Better Self-Consistency.** Self-consistency (Wang et al., 2022) improves reasoning by sampling multiple paths and aggregate together. We combine our method with self-consistency by applying rerouting optimization, then generating multiple reasoning paths with optimized routing decisions. For each sample, we generate 3 reasoning paths. For MMLU, MATH500, and GSM8K, we use majority voting to determine the final accuracy. For code generation tasks (MBPP, HumanEval), we report pass@3 scores. shows substantial improvements with an average 3 percentage point gain over self-consistency alone. We hypothesize that our rerouting framework generates higher-quality reasoning chains by selecting more appropriate experts, and when self-consistency aggregates these improved paths, the voting mechanism amplifies the benefits.

**Table 4:** Performance comparison of individual Test-Time methods versus combined approaches.

| Method | HumanEval | MBPP | GSM8K | MATH500 | MMLU | Average |
|---|---|---|---|---|---|---|
| Baseline | 50.60 | 58.37 | 72.10 | 22.60 | 50.77 | 50.89 |
| Ours | **54.26** | **62.65** | **73.62** | **25.00** | **52.40** | **53.59** |
| *In-Context Learning* | | | | | | |
| ICL (3-shot) | 52.44 | 56.81 | 71.10 | 21.60 | 44.33 | 49.26 |
| **ICL + Ours** | **53.05** | **62.65** | **76.00** | **27.00** | **46.53** | **53.05** |
| *Self-Consistency* | | | | | | |
| Self-Consistency (3) | 51.02 | 70.04 | 75.28 | 26.20 | 51.87 | 54.88 |
| **Self-Consistency (3) + Ours** | **55.08** | **71.21** | **77.54** | **27.40** | **54.20** | **57.09** |

## 6.3 EFFICIENCY ANALYSIS

Beyond performance improvements, we also compare the computational cost of our method with other online approaches to adapt models. Table 3 reports the results on the HumanEval task with the

**Table 5:** Performance of our method versus the baseline on AIME datasets using the *GPT-OSS-20b* model. Optimizing rerouting is especially noticeable at improving model certainty, as shown by improved performance at lower pass@k and improved majority voting, even one challenging benchmarks like AIME.

| Data / Method | Pass@k | | | Maj@k | | | Average |
|---|---|---|---|---|---|---|---|
| | 2 | 4 | 8 | 2 | 4 | 8 | |
| *AIME25* | | | | | | | |
| Baseline | 83.21 | 87.90 | 90.00 | 65.36 | 71.76 | 76.67 | 74.29 |
| **Rewiring (Ours)** | **83.81** | 86.19 | 86.67 | **67.50** | **76.67** | **83.33** | **75.65** |
| *AIME24* | | | | | | | |
| Baseline | 78.57 | 83.76 | 86.67 | 60.60 | 67.48 | 70.00 | 69.58 |
| **Rewiring (Ours)** | **81.67** | **84.95** | 86.67 | **65.83** | **72.57** | **80.00** | **73.75** |

DeepSeek model, showing the average total FLOPs and inference time per sample across different methods. As shown in Table 3, our method achieves notable computational savings over most baselines. Although it requires more computation than the vanilla baseline, it remains substantially more efficient than other test-time techniques, using 1.3× fewer FLOPs than C3PO, 1.6× fewer than ICL (5-shot), and comparable to ICL (3-shot) and Self-Consistency (3). Despite the extra routing operations, our method maintains a competitive inference time of 20.12s, indicating that the overhead of online adaptation is modest while still delivering improved performance.

Conceptually, our method requires $n$ additional prefill passes on already generated text every $m$ tokens, which, given the ease of parallelization of prefill is a manageable compute increase. In this work, we implement this optimization in a straightforward manner, but a production implementation could be significantly faster by incorporating the optimization into disaggregated-prefill systems, or timing the MoE rewiring event with low-load timespans, for example when waiting for a user to respond. The actual routing changes are self-contained in routing parameters $\delta$ (shaped as number of experts by number of layers), so that on-device storage is feasible, even for many separate conversations.

## 6.4 EXTENSION TO LONG-REASONING TASKS

To evaluate the Generalizability of our approach to modern reasoning models, we also test GPT-OSS-20B (Agarwal et al., 2025). We evaluate on the AIME benchmark (MAA Committees), a challenging mathematics competition dataset that requires sophisticated multi-step derivations and mathematical reasoning. This setting allows us to assess whether our online rerouting method can improve performance on the demanding reasoning tasks that represent the current frontier of AI capabilities. As shown in Table 5, our method improves performance on both AIME datasets, achieving higher average correctness (75.65% vs. 74.29% on AIME25; 73.75% vs. 69.58% on AIME24). The primary improvements occur in Maj@k metrics rather than Pass@k, suggesting our routing optimization enhances reasoning consistency rather than solution diversity.

## 6.5 ROBUSTNESS TO CONTEXT SHIFTS IN MULTI-TURN SCENARIOS.

In real-world applications, MoE models often encounter *multi-turn conversations* where contexts shift dramatically between different topics or tasks. To assess the robustness of our method under realistic multi-turn contexts, we simulate context shifts by prepending few-shot examples from different domains before the target task. Figure 4 presents results on HumanEval and Math500 using DeepSeek-V2-Lite, under two conditions: (1) **Aligned-task**, where few-shot examples are from the same task domain, and (2) **Shifted-task**, where examples are drawn from unrelated domains (e.g., MATH and MMLU for code tasks) to induce *cross-domain shifts*.

Across both benchmarks, our method (Ours-Shifted, Ours-Aligned) consistently outperforms the baselines (Base-Shifted, Base-Aligned). On HumanEval, all methods benefit from more-shot examples, but our approach shows a stable improvement. In contrast, baseline gains remain modest, particularly in the Shifted-task setting, where performance fluctuates. On Math500, our method maintains robust and consistent results across both domains, while baselines exhibit limited or even

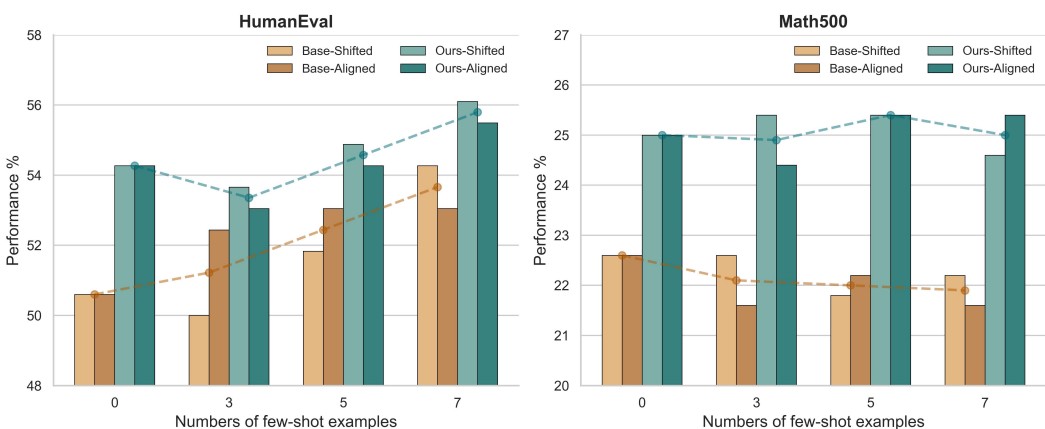

**Figure 4:** Performance of our method versus the baseline across different few-shot examples under shifted and aligned task contexts.

inconsistent improvements. These findings highlight the robustness and scalability of our test-time rerouting strategy in handling diverse contextual information

# 7 CONCLUSION

In this work, we introduce a novel test-time rerouting approach that enables MoE models to dynamically adapt expert selection on the fly, without requiring external data or costly retrieval. The method alternates between routing optimization and steered generation, forming a feedback loop that progressively improves expert selection. To reduce computational overhead, we employ lightweight additive vectors that update only the logits of selected routers. Extensive experiments show that our approach effectively compensates for the inherent imperfections in MoE routing, yielding consistent gains across multiple benchmarks (up to 6.7% on code generation) with 1.6× fewer FLOPs than few-shot methods, while maintaining robustness to context shifts. As a plug-and-play regularization strategy, the method flexibly combines with complementary techniques (e.g., Self-Consistency) to further amplify the benefits. More importantly, by introducing a new dimension of plasticity into MoEs, it opens the door to deployment-time adaptation and points toward practical continual self-regulation in MoE models.

## ETHICS STATEMENT

In this work, we carefully ensure that all methods and experimental protocols conform to established ethical guidelines. Our investigation centers on layer pruning as a strategy to improve the efficiency of LLMs and to lower computational demands, contributing to more sustainable AI practices. In addition, every model and dataset employed in this research is obtained from openly accessible sources, guaranteeing respect for intellectual property and protection of personal privacy. Apart from the models used as experimental subjects (OLMoE, Qwen1.5- MoE, DeepSeek-V2-Lite, and GPT-OSS-20B), we also utilized LLMs as writing assistants, as detailed in Section A. All uses of LLMs in this work comply with the ICLR Code of Ethics.

## REPRODUCIBILITY STATEMENT

We made several efforts to ensure reproducibility. First, we provide detailed experimental settings and hyperparameters used throughout this paper in Section 4, Appendix B, and Section 6.3, and report all evaluation metrics in Section 5.1 and Section 5.2. Second, our code will be submitted with the paper, accompanied by detailed usage instructions and scripts to reproduce all reported results.

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

## A  THE USE OF LARGE LANGUAGE MODELS

We used large language models solely as a general-purpose writing aid to help improve the clarity and readability of the text, and to suggest minor wording improvements. The LLMs did not contribute to the research ideation, experimental design, analysis, or interpretation of results. All technical content, experiments, and conclusions presented in this paper are entirely the work of the authors.

## B  EXPERIMENTAL SETTINGS

### B.1  BENCHMARKS

**MMLU-redux** is a manually re-annotated subset of the original MMLU benchmark designed to address quality issues in the dataset. The dataset contains 3,000 questions across 30 MMLU subjects (100 questions per subject), with expert annotators identifying and categorizing various types of errors using a comprehensive error taxonomy. These errors include issues such as bad question clarity, unclear options, no correct answers, multiple correct answers, and wrong ground truth labels. MMLU-Redux provides a more reliable evaluation standard by filtering out problematic questions and offering corrected annotations where possible.

**HumanEval** is a benchmark dataset for evaluating code generation capabilities of large language models. The dataset consists of 164 hand-crafted programming problems, each including a function signature, docstring, body, and several unit tests (averaging 7.7 tests per problem). These challenges

assess language comprehension, algorithms, and simple mathematics, with difficulty comparable to simple software interview questions.

**MBPP** is a code generation benchmark consisting of around 1,000 crowd-sourced Python programming problems designed to be solvable by entry-level programmers. Each problem includes a task description, code solution, and 3 automated test cases, covering programming fundamentals and standard library functionality. The dataset provides two versions: a full version with 974 problems and a sanitized version with 427 problems. The sanitized split underwent a second round of annotation to improve task descriptions, addressing issues where the original descriptions might not be sufficiently expressive to solve the tasks. This hand-verified subset provides higher-quality problem statements for more reliable evaluation of code generation models. In this paper, we use the sanitized version.

**GSM8K** is a dataset of 8,500 high-quality, linguistically diverse grade school math word problems created by human problem writers. The dataset is segmented into 7,473 training problems and 1,319 test problems. Each problem takes between 2 and 8 steps to solve using basic arithmetic operations, with problems designed so that a bright middle school student should be able to solve every problem. Solutions are provided in natural language format rather than pure mathematical expressions, offering insight into multi-step reasoning processes.

### B.2 BASELINES

**C3PO** We adopt C3PO (Critical-Layer, Core-Expert, Collaborative Pathway Optimization) (Li et al., 2025a) as our primary baseline method. C3PO is a test-time optimization approach designed to address the suboptimal expert pathway selection problem in Mixture of Experts (MoE) large language models. The method operates on the observation that end-to-end trained routers often produce inefficient pathways for challenging or out-of-distribution samples, leading to degraded performance on diverse downstream tasks.

The core idea of C3PO is to dynamically re-mix expert pathways during inference by leveraging successful routing patterns from a reference set. Given a reference set of $m$ samples $\{(x_i, y_i)\}_{i=1}^m$ with their corresponding expert pathway matrices $\{\omega_i\}_{i=1}^m$ (where each $\omega_i \in \mathbb{R}^{L \times E}$ with $L$ layers and $E$ experts) on which the model makes correct predictions, C3PO aims to find an improved pathway matrix $\omega$ for a new test sample $x$ that leads to more accurate outputs.

Among the three optimization strategies proposed by C3PO (gradient descent, kernel regression, and mode finding), we implement the **kernel regression** approach in our experiments. This method estimates optimal expert pathways by computing a weighted average of neighbors' pathway matrices:

$$\hat{\omega} = \frac{\sum_{i \in \mathcal{N}(x)} K(x_i, x) \omega_i}{\sum_{i \in \mathcal{N}(x)} K(x_i, x)} \tag{8}$$

where $K(\cdot, \cdot)$ is a kernel function measuring sample similarity, and $\mathcal{N}(x)$ denotes the neighborhood of $x$ in the reference set. The final pathway is obtained through interpolation:

$$\omega \leftarrow \alpha \omega + (1 - \alpha)\hat{\omega} \tag{9}$$

where $\alpha$ is optimally chosen to minimize the loss function.

For our comparative evaluation, we construct reference sets for each benchmark as follows:

**HumanEval** and **MBPP**, we use the MBPP validation set; for **GSM8K** and **MATH500**, we utilize the GSM8K training set; and for **MMLU**, we employ the MMLU training set. From each source, we randomly sample 100 instances where the model produces the correct predictions, ensuring high-quality pathway references for the optimization process.

**In-Context Learning** In-Context Learning (ICL) is a fundamental capability of large language models that enables them to adapt to new tasks by using a few demonstration examples provided in the input prompt, without requiring parameter updates (Wang et al., 2022; Wei et al., 2022).

In our experimental setup, we implement ICL as a baseline across all evaluation benchmarks. The few-shot examples are carefully selected from relevant datasets to ensure domain alignment and high-quality demonstrations:

- **HumanEval** and **MBPP**: We sample 3-5 examples from the MBPP prompt collection, which provides well-crafted code generation examples with clear problem descriptions and corresponding Python solutions.

- **GSM8K**: We utilize 3-5 examples from the GSM8K training set, featuring step-by-step mathematical reasoning demonstrations that guide the model through problem-solving processes.

- **MATH500**: We sample 3-5 examples from the HYDRA-Math dataset, which offers high-quality mathematical problem-solution pairs covering various difficulty levels and mathematical domains.

- **MMLU**: We select 3-5 examples from the MMLU validation set, ensuring coverage of diverse knowledge domains while maintaining format consistency for multiple-choice questions.

For each benchmark, the few-shot examples are randomly sampled from their respective source datasets and prepended to each test query. This approach provides the model with task-specific context while maintaining consistency across different evaluation scenarios. The number of examples (3-5) is chosen to balance between providing sufficient context and avoiding excessive prompt length that might degrade model performance.

### B.3 MODEL SELECTION

**OLMoE** is a fully open-source MoE language model developed by the Allen Institute for AI and Contextual AI (Muennighoff et al., 2024). The model employs a decoder-only transformer architecture with 1 billion active and 7 billion total parameters. Each MoE layer contains 64 experts, of which 8 are activated per input token through a learned router network. This sparse activation mechanism enables computational efficiency similar to dense 1B parameter models while leveraging the full 7B parameter capacity.

In our experiments, we utilize OLMoE-1B-7B as the base model to evaluate expert pathway optimization strategies across diverse benchmarks.

**DeepSeek-V2-Lite** is a smaller variant of the DeepSeek-V2 model family, developed by DeepSeek-AI (Liu et al., 2024). The model employs innovative architectures, including Multi-head Latent Attention (MLA) and DeepSeekMoE, with 15.7 billion total parameters and 2.4 billion activated parameters per token. DeepSeek-V2-Lite has 27 layers with a hidden dimension of 2048, utilizing MLA with 16 attention heads, where each head has a dimension of 128.

The model adopts the DeepSeekMoE architecture, where all feed-forward networks except the first layer are replaced with MoE layers. Each MoE layer consists of 2 shared experts and 64 routed experts, with 6 experts activated for each token. This sparse activation mechanism enables computational efficiency while maintaining strong performance across diverse tasks.

In our experiments, we use DeepSeek-V2-Lite to evaluate expert pathway optimization strategies, leveraging its balance between model capacity and computational efficiency for comprehensive benchmark evaluation.

**Qwen1.5-MoE** is developed by the Qwen team (Team, 2024). The model is upcycled from the dense Qwen-1.8B model, featuring 14.3 billion total parameters with 2.7 billion activated parameters during runtime. Despite using only 2.7B active parameters, the model achieves comparable performance to Qwen1.5-7B while requiring 75% fewer training resources and demonstrating 1.74x faster inference speed.

The model employs a fine-grained expert architecture with 64 experts total, consisting of 4 shared experts that are always activated and 60 routing experts with 4 activated per token. This configuration represents an 8-fold increase in expert count compared to conventional MoE setups, enabling higher model capacity without proportional parameter increases. The fine-grained expert design

partitions a single FFN into multiple segments, each serving as an individual expert, allowing for more specialized knowledge representation.

In our experiments, we utilize Qwen1.5-MoE to evaluate expert pathway optimization strategies, leveraging its efficient architecture that balances performance and computational cost across diverse evaluation benchmarks.

## C  METHOD DETAILS

### C.1  TEST-TIME MoE REROUTING

We elaborate the rerouting algorithmic pipeline in Algorithm 1, which details the step-by-step procedure for adaptive expert selection and pathway calibration at test time.

### C.2  PATHWAY DIFFERENCES

We first examine how expert pathways change after rerouting using edit distance following (Li et al., 2025b). For input $x_i$, we define its pathway $s_i$ as the ordered sequence of selected experts across $L$ MoE layers as $s_i = \text{concat}(e_1^{(i)}, e_2^{(i)}, \ldots, e_L^{(i)})$, where $e_\ell^{(i)}$ represents expert indices at layer $\ell$ as comma-separated strings (e.g., '3,1,5'), joined across layers with hyphens. We quantify pathway differences using Levenshtein edit distance as $D_{\text{path}}(s_i, s_j) = \text{EditDistance}(s_i, s_j)$. This captures mismatches in expert selection, and pathway shifts.

## D  ADDITIONAL ANALYSIS

### D.1  LAYER-WISE CONFIDENCE DISTRIBUTIONS ACROSS DIFFERENT TASKS

We visualize the confidence distributions across different tasks using DeepSeek-V2-lite as an example. As illustrated in Figure 5, activation patterns across experts and layers are highly task-specific. For instance, math tasks (Math500) and code tasks (HumanEval) exhibit distinct confidence patterns across layers, with math tasks showing higher confidence in middle layers (layers 7-14) while code tasks demonstrate more distributed confidence patterns with peaks in later layers (layers 15-18). Notably, both tasks show generally higher confidence concentrated in the middle-to-later layers compared to early layers.

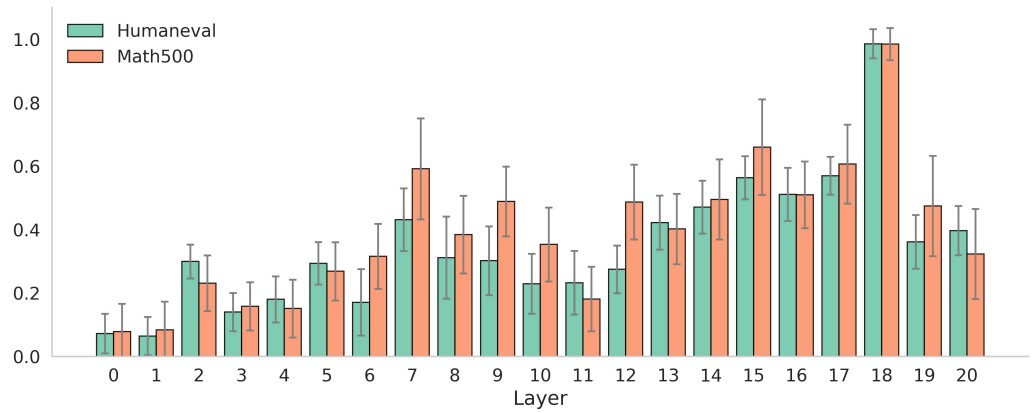

**Figure 5:** Layer-wise confidence distributions across different tasks in DeepSeek-V2-lite-MoE

### D.2  SENSITIVITY ANALYSIS OF HYPERPARAMETERS

**Analysis of Optimization Interval.** As shown in Figure 6, we investigate the impact of optimization interval $m$ (measured in tokens) on performance across five benchmarks. Our test-time rerouting

method consistently outperforms the baseline across all interval settings, demonstrating robustness to this hyperparameter. The results reveal that intervals of 128-160 tokens achieve optimal performance, effectively balancing routing adaptability with computational efficiency.

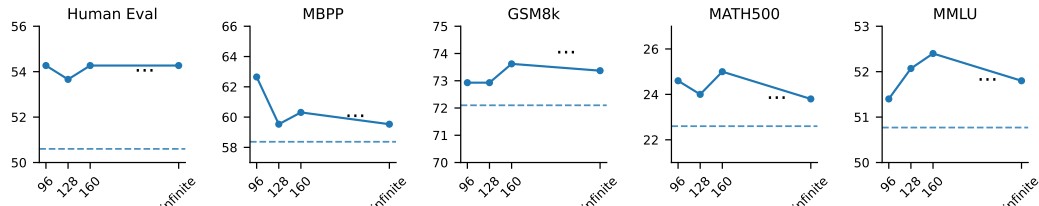

**Figure 6:** Effect of optimization interval on performance across five benchmarks. The dashed line represents baseline results without rerouting. Our method consistently outperforms the baseline, with intervals of 128-160 tokens achieving optimal performance.

**Analysis of Optimization Steps.** We conduct experiments using the DeepSeek V2 model, evaluating its performance on the HumanEval across different optimization step settings. Table 6 presents the performance sensitivity to the number of optimization steps $T$ during test-time adaptation. The results show that performance peaks at $T = 4$ (54.88%), with $T = 3\text{-}5$ forming an optimal range (53.83%-54.88%). This sweet spot allows sufficient adaptation without over-fitting to individual test instances.

Beyond $T = 5$, performance gradually declines, suggesting over-adaptation where the model begins to fit noise rather than meaningful patterns in the test data. Interestingly, even with suboptimal step counts, our method maintains consistent improvements across $T = 3\text{-}7$ (all $> 52\%$), demonstrating robustness to this hyperparameter. The relatively flat performance curve within the optimal range indicates that practitioners can select values in this region without significant performance penalties, making the method practical for deployment.

**Table 6:** Performance (%) across different optimization steps $T$ on HumanEval benchmark. The optimal range is $T = 3\text{-}5$, with peak performance at $T = 4$.

| T | 1 | 2 | 3 | 4 | 5 |
|---|---|---|---|---|---|
| **Performance** | 50.60 | 50.17 | 53.83 | **54.88** | 54.27 |
| **T** | 6 | 7 | 8 | 9 | 10 |
| **Performance** | 52.95 | 53.83 | 51.56 | 51.83 | 51.83 |

## D.3 IMPACT ON SAMPLE DIVERSITY IN PARALLEL GENERATION.

**Table 7:** Diversity evaluation results comparison

| Metric | Baseline | Ours |
|---|---|---|
| Cosine Div. | $0.390 \pm 0.157$ | $0.380 \pm 0.150$ |
| Semantic Div. | $0.012 \pm 0.007$ | $0.012 \pm 0.008$ |

**Impact on Sample Diversity in Parallel Generation.** While our online adaptation method improves routing efficiency, we investigate whether the adapted routing decisions might reduce sample diversity when generating multiple sequences in parallel. To quantify this effect, we evaluate the diversity of generated samples using two complementary metrics: semantic diversity based on Code-BERT embeddings (Feng et al., 2020), which captures functional similarities between code snippets, and TF-IDF-based cosine diversity, which measures lexical variation. We used the DeepSeek MoE model with default settings to generate 10 samples on the HumanEval task. Table 7 shows that our method maintains comparable diversity to the baseline across both metrics. Cosine diversity scores are nearly identical ($0.380 \pm 0.150$ vs. $0.390 \pm 0.157$), as are semantic diversity scores ($0.012 \pm 0.008$ vs. $0.012 \pm 0.007$). These results demonstrate that our online adaptation preserves sample diversity while improving routing efficiency.

---

**Algorithm 1** Test-Time MoE Rerouting

---

**Input:** Pre-trained MoE model $\mathcal{M}$; input prompt $\mathbf{x} = (x_1, \ldots, x_n)$; optimization steps $n$; generation interval $m$; optimizer $\mathcal{O}$.
**Output:** Generated response $\mathbf{y}$.

**// Parameter Initialization**
Initialize $\boldsymbol{\delta}^{(l)} = \mathbf{0} \in \mathbb{R}^N$ for all layers $l \in \mathcal{L}$;

**// Phase 1: In-Context Routing Optimization**
$\mathbf{x}_{\text{current}} = \mathbf{x}, T = |\mathbf{x}|$;
Select layers $\mathcal{S} = \text{TopK}(\{C^{(l)}\}_{l=1}^L, r)$ or compute soft weights $\{w_l\}_{l=1}^L$;
**for** $i = 1$ to $n$ **do**
Compute loss $\mathcal{L}(\{\boldsymbol{\delta}^{(l)}\}_{l=1}^L) = -\sum_{j=1}^{T-1} \log p(x_{j+1} \mid x_{1:j}, \{\boldsymbol{\delta}^{(l)}\}_{l=1}^L)$;
**if** hard selection **then**
**for** $l \in \mathcal{S}$ **do**
$\boldsymbol{\delta}^{(l)} \leftarrow \mathcal{O}(\boldsymbol{\delta}^{(l)}, \nabla_{\boldsymbol{\delta}^{(l)}} \mathcal{L})$;
**end for**
**else** **//** soft weighting
**for** $l = 1$ to $L$ **do**
$\boldsymbol{\delta}^{(l)} \leftarrow \mathcal{O}(\boldsymbol{\delta}^{(l)}, w_l \nabla_{\boldsymbol{\delta}^{(l)}} \mathcal{L})$;
**end for**
**end if**
**end for**

**// Phase 2: Steered Generation with Periodic Re-optimization**
Initialize $\mathbf{y} = ()$;
**repeat**
**//** *Generate m tokens using optimized routing parameters*
**for** $k = 1$ to $m$ **do**
Generate $x_{\text{next}} \sim p(\cdot \mid \mathbf{x}_{\text{current}}, \{\boldsymbol{\delta}^{(l)}\}_{l=1}^L)$;
Append $x_{\text{next}}$ to $\mathbf{y}$ and $\mathbf{x}_{\text{current}}$;
**end for**
**//** *Re-optimize with extended context*
$T = |\mathbf{x}_{\text{current}}|$;
Select layers $\mathcal{S} = \text{TopK}(\{C^{(l)}\}_{l=1}^L, r)$ or compute soft weights $\{w_l\}_{l=1}^L$;
**for** $i = 1$ to $n$ **do**
Compute loss $\mathcal{L}(\{\boldsymbol{\delta}^{(l)}\}_{l=1}^L) = -\sum_{j=1}^{T-1} \log p(x_{\text{current},j+1} \mid x_{\text{current},1:j}, \{\boldsymbol{\delta}^{(l)}\}_{l=1}^L)$;
**if** hard selection **then**
**for** $l \in \mathcal{S}$ **do**
$\boldsymbol{\delta}^{(l)} \leftarrow \mathcal{O}(\boldsymbol{\delta}^{(l)}, \nabla_{\boldsymbol{\delta}^{(l)}} \mathcal{L})$;
**end for**
**else** **//** soft weighting
**for** $l = 1$ to $L$ **do**
$\boldsymbol{\delta}^{(l)} \leftarrow \mathcal{O}(\boldsymbol{\delta}^{(l)}, w_l \nabla_{\boldsymbol{\delta}^{(l)}} \mathcal{L})$;
**end for**
**end if**
**end for**
**until** end-of-sequence or max length reached;

**return** $\mathbf{y}$;

---

### D.4 IMPACT ON INTERNAL CONFIDENCE IN GENERATION

We further investigate how our method influences the internal confidence during generation. The central hypothesis is that model uncertainty, quantified using information-theoretic measures such as entropy or softmax margins, can serve as a reliable indicator of generation reliability.

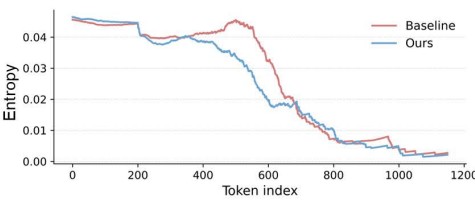 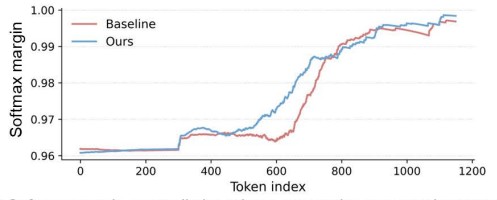

a) Entropy on predictive tokens across the generated sequence.  b) Softmax margin on predictive tokens across the generated sequence.

**Figure 7:** Confidence metrics across token positions during sequence generation.

Let $P(y_t|y_{<t}, x)$ denote the output distribution of the model at time step $t$, conditioned on previously decoded tokens $y_{<t}$ and the input sequence $x$. For each position $t$, a scalar confidence score $C_t$ is computed that reflects the model's certainty on its prediction.

Here, we use two kinds of metrics to estimate the confidence score: (1) entropy-based confidence and (2) softmax margin confidence.

**Entropy-based Confidence:** Entropy serves as a fundamental measure of uncertainty in probabilistic models. For token-level generating, the entropy of the model's predictive distribution is given by: $C_t^{(ent)} = -\sum_{y \in V} P(y|y_{<t}, x) \times \log P(y|y_{<t}, x)$. Here, $V$ denotes the output vocabulary, and $P(\cdot|y_{<t}, x)$ is the model's probability distribution over the next token given the previous context and input. A lower value of $C_t^{(ent)}$ corresponds to a sharper (more peaked) distribution, implying higher confidence in the predicted token. Conversely, a higher entropy indicates greater uncertainty and suggests a more conservative generation.

**Softmax margin confidence:** Softmax margin confidence captures the model's certainty by evaluating the probability gap between the top two predicted tokens after applying the softmax function. At each decoding step $t$, let $y^{(1)}$ and $y^{(2)}$ denote the top-1 and top-2 predicted tokens by the model, and let their softmax probabilities be: $P^{(1)} = P(y^{(1)}|y_{<t}, x)$ and $P^{(2)} = P(y^{(2)}|y_{<t}, x)$. The softmax margin confidence is then computed as: $C_t^{(soft)} = P^{(1)} - P^{(2)}$.

The softmax margin confidence reflects how decisively the model prefers its most likely token over the next best alternative, judged by the normalized probability distribution. A higher value of $C_t^{(soft)}$ implies strong confidence in the top choice, whereas smaller values indicate ambiguity, with similar likelihoods assigned to multiple candidates.

As shown in Figure 7, compared to the baseline, our method demonstrates notable improvements in confidence characteristics across the generated sequence. In Figure 7(a), the entropy-based confidence reveals that our approach achieves lower entropy values, particularly in the middle and later stages of generation (token index 400-1000), indicating more peaked probability distributions and higher prediction certainty. This suggests that our method produces more confident and decisive token predictions. Meanwhile, Figure 7(b) shows that our method consistently maintains higher softmax margin values throughout the generation process, especially after token index 400, demonstrating a clearer preference for the top predicted token over alternatives. The increased margin between top-1 and top-2 predictions indicates that our method reduces ambiguity and enhances the model's decision confidence. These results collectively demonstrate that our approach enhances the model's internal confidence, leading to more reliable and deterministic outputs.

### D.5   WHAT IF WE OPTIMIZE THE FULL LINEAR ROUTING LAYER?

To validate our design choice of bias-only optimization, we compare it against alternative parameter adaptation strategies. Table 8 presents the results across different optimization approaches on the HumanEval benchmark. Results show that: (1) Full linear layer optimization leads to complete performance collapse, indicating severe instability when optimizing all routing parameters at test time. (2) Low-rank LoRA (rank-1), despite lower capacity than full layer optimization, underperforms even the baseline, demonstrating that **increased capacity does not guarantee better test-time adaptation**. Our bias-only approach achieves the best performance (54.26%) with minimal parameters, effectively balancing adaptation capability and optimization stability.

**Table 8:** Performance comparison of different parameter adaptation strategies on HumanEval benchmark.

| Method | HumanEval (%) |
|---|---|
| Baseline | 50.60 |
| **Ours (bias-only)** | **54.26** |
| Rank-1 LoRA | 46.80 |
| Full linear layer | collapse |

### D.6 PERFORMANCE ANALYSIS ACROSS DIFFERENT SEQUENCE LENGTHS

We conducted additional analysis on HumanEval by stratifying samples based on their sequence lengths and computing accuracy for each bucket. As shown in Table 9, our method demonstrates increasing performance gains as sequence length grows. For short sequences ($< 256$ tokens), both methods achieve perfect accuracy (100.0%). However, as sequences become longer, our method shows progressively larger improvements: +2.9% for 256-384 tokens, +3.6% for 384-512 tokens, and +2.4% for 512-640 tokens. Most notably, for very long sequences exceeding 640 tokens, our method achieves a substantial +13.3% improvement over the baseline (23.3% vs. 10.0%). This trend validates that test-time routing optimization becomes increasingly beneficial as context length grows.

**Table 9:** Performance comparison across different sequence lengths on HumanEval. Our method shows increasing gains on longer sequences.

| Length Range | Ours | Baseline | Gain |
|---|---|---|---|
| $< 256$ | 100.0% | 100.0% | +0.0% |
| 256-384 | 73.5% | 70.6% | +2.9% |
| 384-512 | 61.8% | 58.2% | +3.6% |
| 512-640 | 46.3% | 43.9% | +2.4% |
| $> 640$ | 23.3% | 10.0% | **+13.3%** |

