# OpenReview forum: "Rewiring Experts on the Fly: Continuous Rerouting for Better Online Adaptation in Mixture-of-Expert models"
_ICLR.cc/2026/Conference — Submitted to ICLR 2026_

### Official Review · Reviewer_ufCS · 2025-10-27

**Soundness:** 3
**Presentation:** 3
**Contribution:** 2
**Rating:** 4
**Confidence:** 4

**Summary:**

This paper proposes a data-free, online test-time adaptation framework for MoE LLMs. It continuously optimizes expert routing decisions during text generation by introducing lightweight additive parameters to router logits. To further improve efficiency and stability, the framework updates only a subset of MoE layers determined by confidence-based layer selection, which identifies task-relevant layers.

**Strengths:**

The paper tackles an important problem of test-time adaptation for MoE models and presents an effective solution. The proposed method is data-free, computationally efficient, and easily applicable to existing models. Experiments show consistent performance gains across diverse reasoning and generation tasks.

**Weaknesses:**

(1) The paper lacks theoretical analysis explaining why optimizing routing decisions on the current context leads to better performance on future generation. What guarantees that reducing cross-entropy loss on already-generated tokens improves routing for subsequent reasoning steps? This is particularly concerning since the optimization objective (past context) differs from the actual goal (future generation quality).

(2) The evaluation focuses primarily on relatively short reasoning tasks (coding problems, math word problems). The method's effectiveness on longer-context scenarios (multi-turn dialogues, document summarization, extended reasoning chains) remains unclear. Given that the optimization interval is set to 128 tokens, how does performance scale with generation length? Does the method maintain benefits after generating more tokens, or do cumulative routing adjustments lead to drift?

(3) While the ablation study (Table 2) shows that confidence-based selection outperforms alternatives, the underlying assumption (high-confidence layers are more "important" for adaptation) lacks empirical validation. High confidence could equally indicate that these layers have already converged to good routing decisions and need less adjustment. Do high-confidence layers exhibit more task-relevant expert specialization? Are low-confidence layers fundamentally noisy or processing difficult parts of the task?

**Questions:**

Do the reported hyperparameters (η=0.05, n=5, m=128) transfer across different models and tasks, or does each setting require task-specific tuning?

---

> ### Author Response · Authors · 2025-11-20
>
> Thank you for finding our work efficient, and easily applicable to current MoE models. You can find responses addressing your questions below:
>
> ## Response to theoretical analysis about routing
>
> Theoretically, optimizing based on current context is well-supported as a self-supervised training objective. While one could certainly imagine even more complicated training objectives, for  example using per-token rewards or task-specific process verification, such supervision is not as easy to obtain. It is a key finding of our work, that routing decisions can be optimized based on already generated context. We extensively evaluate this hypothesis empirically, and find to hold for multiple models and all datasets we test.
>
>
> Conceptually, our rerouting optimization is based on the hypothesis that supervised prompt tokens provide a reliable signal for adjusting routing, and that the adapted routing patterns naturally carry over to improve downstream generation. **This assumption is motivated by the fact that language is inherently self-similar, i.e. it exhibits strong word–word associations and co-occurrence structures. As a result, prompt tokens and generated tokens are hypothesized to be highly correlated semantically, which makes routing adaptations learned during the prompt phase naturally transfer to and influence the subsequent generation process.**
>
> Notably, our empirical results show consistent gains across multiple benchmarks and diverse tasks. Furthermore, our analysis in Section 6.1 shows that our method improves routing confidence, and the additional experiment in Section D.4 demonstrates that the impact on token-level uncertainty throughout generation is minimal. Taken together, these findings suggest that the gains of our method primarily arise from the following factors:
>
> 1. **Context optimization finetunes the MoE to the current task context.** During the prompt phase, supervision from the prompt guides the routing network toward experts that best match the task context. These routing patterns then persist and lead to more stable and coherent outputs during the generation of the answer to the prompt.
> 2. **Context optimization decreases Routing errors.** As, we measure, our intervention decreases the entropy of the router layers in the model without strongly decreasing the model output’s entropy. Put simply, the rewiring makes the expert selection more confident, and as such, removes noise from incorrect expert selection, without making the actual outputs of the model overconfident (as one could have conjectured based on the context optimization objective).
>
> Our method enhances generation correctness and consistency through test-time routing adaptation, leading the model to pick “better” experts for the task at hand. Our extensive experiments show that with this mechanism, it provides a practical solution to data-free test-time adaptation.

---

> ### Author Response · Authors · 2025-11-20
>
> ## Response to Effectiveness on Longer-Context Scenarios
>
> Please note that we do extensively evaluate the effectiveness of our approach in long-context scenarios. You can find our evaluations on longer-context scenarios in Sections 6.4 and 6.5 of the submission. We especially want to highlight two interesting long-context findings:
>
> **Multi-turn dialogues with context shifts (Section 6.4):** We simulate multi-turn scenarios by prepending few-shot examples from different domains (MATH, MMLU) before the target task (HumanEval). Results show our method maintains robust performance under cross-domain context shifts, even as context lengths (here via number of few-shot examples) increase.
>
> **Extended reasoning chains (Section 6.5):** We evaluate our approach on  theAIME 24/25 competition mathematics benchmark, which requires multi-step reasoning often exceeding **8k tokens**. Our method achieves consistent improvements (AIME25: 75.65% vs. 74.29%; AIME24: 73.75% vs. 69.58%).
>
> Both experiments demonstrate our method's effectiveness and robustness in long-context settings.
>
>
> ##  Response to empirical validation of layer selection
>
> Please note that we also do ablate the hypothesis that "high-confidence layers are more "important" for adaptation".
> You can find these ablations in Table 2, named 'Ablation Study: Layer Selection Strategies', which includes for example the "Reverse metric" baseline that selects low-confidence layers for optimization. Results show consistent degradation across all benchmarks (worst performance), demonstrating that low-confidence layers are not suitable for test-time optimization.
>
> ## Response to hyperparameter selection details
>
> We apologize for any unclear details. We really do use the same hyperparameters (η=0.05, n=5) across all tasks and models **without tuning**. The only adjustment we made was setting m=128 tokens for math and MMLU tasks, and m=96 tokens for code generation tasks, since code generation produces more concise outputs, as detailed in Section D.2 of the submission. Our method consistently outperforms the baseline across all interval settings, while task-specific tuning might yield additional gains or minor variations; we provided corresponding analysis of hyperparameter sensitivity in Appendix D.2.
>
> Thank you for your detailed comments. We hope that our clarifications have addressed your concerns. Please don’t hesitate to reach out with any further questions. We would be happy to discuss further.

---

> > ### Author Response · Authors · 2025-11-26
> >
> > Thank you again for your thoughtful review. We wanted to briefly follow up to ask whether our responses address your questions, or if there is anything else we can clarify.

---

> > ### Comment · Reviewer_ufCS · 2025-11-27
> > **Response to Authors**
> >
> > Thank you for your detailed response. Most of my concerns are now addressed. However, my central concerns remains unclear:
> >
> > 1.Theoretical Gap: Author's explanation relies on the "self-similarity" of language. But this remains an intuitive hypothesis. It does not rigorously explain why minimizing loss on the past context guarantees better routing for future steps, especially when the optimization objective differs from the generation goal. For example, in reasoning tasks, the input (problem) is often very different from the output (solution steps).
> >
> > 2.Mechanism: The explanation for why modify high-confidence layers is still unclear. If the router is already certain (high confidence), why does perturbing it improve performance? Does this imply the pre-trained router is often "confidently wrong"?

---

> > > ### Author Response · Authors · 2025-11-27
> > >
> > > Thank you for your reply. Here is our further clarification.
> > > ## Theoretical Gap
> > > We acknowledge that we do not provide strict theoretical guarantees for why optimizing on past context improves future routing. **However, our extensive experimental results demonstrate the effectiveness of this approach, which we also find to be a surprising and noteworthy finding.**
> > > Regarding the concern that "the input (problem) is often very different from the output (solution steps)," we argue that this difference is superficial. The problem and solution steps are inherently connected through logical reasoning chains; this is precisely the principle underlying how language models work. A well-formed solution to a given problem should exhibit lower perplexity because it maintains coherent semantic and logical flow with the problem context. Conversely, irrelevant or incorrect content would result in higher perplexity. That the model learns to assign higher probability (lower perplexity) to contextually appropriate continuations is central to next-token prediction, and not just our intuition.
> > > In our test-time optimization framework, by minimizing loss on the observed context, we are essentially calibrating the router to favor expert pathways that better capture the task-specific patterns and reasoning structures present in the current instance. While the input and output may appear different in surface form, they share underlying semantic and logical dependencies that the router can learn to leverage through our optimization approach.
> > > We recognize that establishing formal theoretical guarantees remains an important direction for future work. Nevertheless, the consistent improvements across diverse tasks and datasets provide strong empirical evidence that optimizing routing based on context alignment is interestingly a valid and effective strategy.
> > >
> > >
> > >
> > > ## Mechanism
> > > We want to clarify that we do not imply the pre-trained router is "confidently wrong." Rather, we acknowledge that the pre-trained router is not always optimal, which is precisely why test-time methods are effective. Like all test-time approaches, we believe our method helps the router better adapt to the current context.
> > > Regarding what the router actually changes, we provide analysis showing that even high-confidence layers have room for optimization. Improvements can come from two sources: (1) pathway changes where the router selects different experts (but minor), or (2) weight adjustments that emphasize certain experts even when the pathway remains unchanged. Both mechanisms can lead to further gains.
> > > We acknowledge that this finding may seem counterintuitive. However, these high-confidence layers demonstrate their importance for the specific task at hand. We do not deny that optimizing low-confidence layers might yield larger improvements, which may be more suitable for fine-tuning scenarios. However, in test-time settings with limited data, such approaches may not be as effective.
> > > High-confidence layers, despite their confidence, do possess room for further optimization. While these improvements may be modest, they are sufficient and meaningful in test-time scenarios. Our method does not aim to fundamentally resolve all router certainty issues. Instead, it emphasizes the important components on top of the existing foundation. This finding aligns with our former work [1].
> > >
> > > [1] Li Z, Li Z, Zhou T. C3po: Critical-layer, core-expert, collaborative pathway optimization for test-time expert re-mixing[J]. arXiv preprint arXiv:2504.07964, 2025.

---

### Official Review · Reviewer_Vpis · 2025-10-30

**Soundness:** 3
**Presentation:** 3
**Contribution:** 2
**Rating:** 6
**Confidence:** 3

**Summary:**

The paper addresses a fundamental challenge in Mixture-of-Experts (MoE) models: **suboptimal routing decisions during inference**. MoE models achieve efficiency by selectively activating only a subset of experts for each token, but the quality of these routing decisions directly impacts performance. The authors draw an analogy to neuroplasticity in the human brain, arguing that MoEs need similar adaptive mechanisms during deployment.

Challenges addresseed in this work in regards to routing:

- Routers are simple linear functions that must approximate the anticipated utility of activating each expert
- Distribution shifts between training and test data lead to poor routing choices
- During standard inference, there's no mechanism to reinforce successful routing or reduce routing to unhelpful experts
- Existing test-time adaptation methods require external reference data, limiting practicality

### Key Contributions and Methodology

The paper introduces a **data-free, online test-time rerouting framework** with three main components:

1. Router Logits Modification (§ 3.2)
    - Introduces lightweight additive parameter vectors $δ^{(l)}$ for each MoE layer
    - Modifies router logits: $z̃^{(l)} = z^{(l)} + δ^{(l)}$
    - These vectors **steer expert selection** without modifying the underlying model weights
2. Dynamic Layer Selection (§ 3.3)

    The framework selectively updates only high-confidence layers rather than all layers:

    - **Confidence metric**: $C_i = -\frac{1}{k} \sum_{j=1}^{k} \log p_{i,j}$ (confidence is calculated as the negative log probability of selected experts; higher values indicate more decisive routing)
    - **Two strategies**:
        - Hard selection: Choose top-r proportion ($S_t = \text{TopK}(\{C_i\}_{i=1}^L, r)$) of highest confidence layers
        - Soft weighting: Apply confidence-based weights to gradient updates
3. Two-Phase Optimization Procedure (§ 3.4)

    The method alternates between:

    - **Phase 1 - In-Context Routing Optimization**: Uses the current context as self-supervised training data, optimizing  to minimize cross-entropy loss over n=5 iterations
    - **Phase 2 - Steered Generation**: Generates m=128 tokens with optimized routing, then returns to Phase 1 with extended context

**Strengths:**

- **Data-free self-supervised adaptation**: The method treats each input as its own training sample, computing cross-entropy loss on the current context (prompt + generated text) to optimize routing parameters $\delta^{(l)}$ without any external data or retrieval overhead. This achieves better performance than baselines that use external data
- **Self-Supervised Routing Refinement**: The framework treats each input prompt as a self-supervised learning opportunity for dynamic expert selection. During inference, the method alternates between two phases: In-Context Routing Optimization, where the current context itself becomes the training data for computing gradient updates to router logits, and Steered Generation, where text is generated using these optimized routing decisions. This creates a dynamic feedback loop - as the model generates text, that very text provides the supervision signal for improving subsequent routing choices. The model continuously refines its understanding of what the task requires based on its own generation progress, leading to increasingly informed expert routing decisions as generation proceeds.
- **Dynamic continuous optimization during generation**: Unlike static approaches, the method alternates between optimization phases (updating $\delta^{(l)}$) and generation phases at 128-token intervals, allowing routing to adapt as the model's understanding of the task evolves. This creates feedback loop where generation quality informs subsequent routing.
- Mechanistic insights that validate the approach:
    - Edit distance analysis shows task-specific pathway modifications concentrated in deeper layers
    - Expert utilization heatmaps reveal strategic redistribution rather than uniform changes
    - Confidence-based layer selection outperforms other form of updates
- The paper follows a natural progression from problem identification through methodology, experiments, and analysis, making it easy to follow the authors' reasoning and working.

**Weaknesses:**

- The paper optimizes routing by adding bias vectors to router logits: $\tilde{z}^{(l)} = z^{(l)} + \delta^{(l)}$ where $z^{(l)} = W_r^{(l)} h^{(l)}$. However, this approach doesn't address the inherent constraint that MoE routing remains a linear function of the hidden state. The modification merely shifts decision boundaries in the existing linear space.

    The relatively small improvements despite optimizing on the exact test context suggest that the linear routing architecture itself **may** be the limiting factor.

- This paper fundamentally assumes that different experts have learned distinct, task-relevant capabilities worth optimizing routing for. However, it never validates this critical assumption. Recent research has shown that MoE experts often exhibit significant redundancy and lack clear specialization, which raises questions about what this routing optimization actually achieves.

    There is no clear anaylsis on the following:

    - Whether experts are meaningfully specialized
    - What the "preferred" experts actually compute differently
    - If routing changes actually correlate with expert capabilities
- The paper's central approach has a fundamental flaw: it optimizes routing to minimize cross-entropy loss on **already-generated text**, then uses those parameters to generate **new text**. This could create problems like:
    - The routing that best explains existing tokens may be entirely different from routing that generates good future tokens
    - The model is essentially being trained to "retroactively justify" its past decisions rather than improve future ones
    - No evidence is provided that minimizing reconstruction loss on context correlates with better generation quality

**Questions:**

How much performance gain can realistically be achieved by steering the router which is fundamentally a linear layer?

---

> ### Author Response · Authors · 2025-11-20
>
> Thank you for for your detailed feedback and support for our work. We’ll answer your questions below.
>
> ## Response to why not optimizing the full linear routing layer and the benefits it could bring
>
> Thank you for this insightful suggestion.
>
> We agree that optimizing the full linear routing layer (W_router ∈ ℝ^(hidden_dim × num_experts)) would provide greater expressive capacity. However, we find such approaches **unsuitable for test-time adaptation scenarios** due to interesting fundamental constraints:
>
> **1. Overfitting risk:** Our data-free test-time adaptation operates on limited context from individual samples. High-dimensional parameter optimization (full weight matrices) on such limited data may create a significant risk of overfitting to sample-specific features that are not predictive of future routing decisions.
>
> **2. Computational efficiency:** Test-time adaption needs to run during real-time inference. Optimizing entire linear layers would increase computational cost and memory requirements.
>
> **3. Optimization stability:** Full layer optimization with limited gradient signals may lead to unstable updates and performance collapse.
>
> Based on your questions we now also validated these constraints empirircally, by conducting ablation experiments that compare the different parameterization strategies:
>
> | Method | HumanEval |
> |--------|-----------|
> | Baseline | 50.60 |
> | **Ours (bias-only)** | **54.26** |
> | Rank-1 LoRA | 46.80 |
> | Full linear layer | collapse |
>
> These results show that (1) Full linear optimization leads to performance collapse due to difficulty optimzing these target (2) Low-rank LoRA (despite lower capacity than full layer) underperforms the baseline (where nothing is optimized), which shows that  **increased capacity does not necssarily lead to better test-time adaptation**.
>
> Our bias-only approach achieves the best performance with, or rather due to its minimal parameter impact. Our approach does not modify how routing decisions are computed from hidden states, it only steers the model by applying slight biases to certain routers.
>
>
>
>
> ## Response to Assumption About Expert Specialization
>
>
>
> **We do not assume that experts are well-specialized.** Our method is agnostic to expert specialization because it directly optimizes routing configurations through end-to-end, gradient-based optimization. The routing parameters are updated solely based on which expert selections reduce the loss in the current context, allowing the model to automatically adjust expert choices without relying on any prior assumption about specialization.
>
> As such, our method does not require experts to be pre-specialized. Rather, we argue that it leverages the fact that adapting routing decisions based on the current context naturally "up-votes" those pathways that are most effective for the current task. This line of thinking is in line with recent work on the interpretability of MoEs, which has also shown that intervening on expert choices can yield precise behavioral control, even in the absence of strong task specialization [1][2]. These observations are consistent with our findings.
>
> [1] Mixture of Tunable Experts--Behavior Modification of DeepSeek-R1 at Inference Time
>
> [2] Two experts are all you need for steering thinking: Reinforcing cognitive effort in moe reasoning models without additional training

---

> ### Author Response · Authors · 2025-11-20
>
> ## Regarding the routing optimization setting
>
> Theoretically, optimizing based on the current context is well-supported as a self-supervised training objective. While one could certainly imagine even more complicated training objectives, for example, using per-token rewards or task-specific process verification, such supervision is not as easy to obtain. It is a key finding of our work, that routing decisions can be optimized based on already generated context. We extensively evaluate this hypothesis empirically, and find that it holds for multiple models and all datasets we test.
>
>
> Conceptually, our rerouting optimization is based on the hypothesis that supervised prompt tokens provide a reliable signal for adjusting routing and that the adapted routing patterns naturally carry over to improve downstream generation. **This assumption is motivated by the fact that language is inherently self-similar, i.e. it exhibits strong word–word associations and co-occurrence structures. As a result, prompt tokens and generated tokens are hypothesized to be highly correlated semantically, which makes routing adaptations learned during the prompt phase naturally transfer to and influence the subsequent generation process.**
>
> Notably, our empirical results show consistent gains across multiple benchmarks and diverse tasks. Furthermore, our analysis in Section 6.1 shows that our method improves routing confidence, and the additional experiment in Section D.4 demonstrates that the impact on token-level uncertainty throughout generation is minimal. Taken together, these findings suggest that the gains of our method primarily arise from the following factors:
>
> 1. **Context optimization finetunes the MoE to the current task context.** During the prompt phase, supervision from the prompt guides the routing network toward experts that best match the task context. These routing patterns then persist and lead to more stable and coherent outputs during the generation of the answer to the prompt.
> 2. **Context optimization decreased router errors.** As, we measure, our intervention decreases the entropy of the router layers in the model without strongly decreasing the model output’s entropy. Put simply, the rewiring makes the expert selection more confident, and as such, removes noise from incorrect expert selection, without making the actual outputs of the model overconfident (as one could have conjectured based on the context optimization objective).
>
> Our method enhances generation correctness and consistency through test-time routing adaptation, leading the model to pick “better” experts for the task at hand. Our extensive experiments show that with this mechanism, it provides a practical solution to data-free test-time adaptation.
>
>
> Thank you for your detailed comments. Please don’t hesitate to reach out with any further questions. We would be happy to discuss further.

---

> > ### Author Response · Authors · 2025-11-26
> >
> > Dear reviewer, we'd like to kindly follow up on our responses to your reviews. We would greatly appreciate it if you could let us know whether our responses address your concerns or if any further clarification is needed.

---

### Official Review · Reviewer_94hV · 2025-11-01

**Soundness:** 1
**Presentation:** 2
**Contribution:** 2
**Rating:** 2
**Confidence:** 4

**Summary:**

This paper proposes a data-free and online test-time framework that continuously adapts MoE routing decisions during text generation without external supervision or data.

**Strengths:**

- The method demonstrates consistent and significant performance gains across multiple, distinct MoE architectures and a diverse set of challenging reasoning benchmarks.
- Excellent Analysis and Ablations. The ablation studies in Section 5.2 thoroughly validate the key design choices: the superiority of confidence-based layer selection over alternatives and the benefit of continuous refinement.

**Weaknesses:**

- Concerns regarding Latency. The paper's claim of a "modest"  computational overhead is a significant understatement. According to Table 3, the proposed method nearly doubles the inference time on HumanEval (20.12s) compared to the baseline (10.71s). Also, the FLOPs for proposed method is 1.96e+12, which is **larger** than 1.93e+12 of ICL(3-shot).

- Lack of sensitivity analysis on optimization steps T.

**Questions:**

In the abstract, this paper claim that existing test-time adaptation methods could potentially address these issues, but they primarily focus on dense models and require access to external data. Also, in the related work, this paper claim that these methods assume accessible training data during deployment and introduce significant retrieval overhead, limiting real-world practicality. It would be better to showcase detailed performance of these methods.

---

> ### Author Response · Authors · 2025-11-20
>
> Thank you for your feedback. We appreciate that you found that our paper contained Excellent Analysis and Ablations. You can find responses addressing your questions as follows:
>
>
> ## Response to Latency
>
>
> Thank you for highlighting this important point. We acknowledge that our phrasing of "modest" overhead requires clarification.
> **Regarding inference time:** The increase in wall-clock time (20.12s) reflects our current (not fully unoptimized) research implementation.
>
> However, it is important to separate current implemntation time from theoretical computational overhead: Conceptually, our method performs **N additional prefill passes every M tokens on already-generated text**, where N is equal to the number of optimization steps, which we set to N=5, and M is 128 in our experiments. Crucially, prefill operations are highly parallelizable, but our implementation does not yet exploit this optimization. A production deployment can substantially reduce this overhead through:
>
> 1. **Disaggregated prefill systems** that parallelize the additional passes, by sending adaptation requests back to prefill nodes which are underutilized during generation
> 2. **Strategic scheduling** of MoE rewiring during low-load periods (e.g. while waiting for user input)
>
> As such, we do not see a fundamental issue with the latency of the proposed adaptation strategy.
>
>
> **Regarding FLOPs:** Our method uses 1.96e+12 FLOPs versus ICL (3-shot)'s 1.93e+12. However, we want to highlight that our approach provides the optimal **performance-efficiency tradeoff**:
>
> - **1.3× fewer FLOPs than C3PO** , while achieving better performance (53.59 vs 48.10 average accuracy across 5 benchmarks in Table 1)
> - **1.6× fewer FLOPs than ICL (5-shot)**, while outperforming it  (53.59 vs 49.08 average accuracy across 5 benchmarks in Table 1)
> - **Comparable FLOPs to ICL (3-shot)** (1.96e+12 vs 1.93e+12), but with higher average accuracy across 5 benchmarks (53.59 vs 49.26 average accuracy in Table 1)
>
> While our method introduces additional computational cost, it does achieves accuracy improvements  with FLOPs comparable to or lower than other test-time adaptation baselines.
>
>
> ## Regarding a sensitivity analysis on optimization steps T.
>
> Thank you bringing this up, we have now included a full ablation of T in our submission:
>
>
> | Optimization Steps T | 1      | 2      | 3      | 4      | 5      | 6      | 7      | 8      | 9 |10|
> |--------|--------|--------|--------|--------|--------|--------|--------|--------|-|-|
> | Performance  | 0.5060 | 0.5017 | 0.5383 | 0.5488 | 0.5427 | 0.5295 | 0.5383 | 0.5156 |0.5183|0.5183|
>
> We conduct experiments using Deepseek-V2-Lite, evaluating on the HumanEval benchmark across different optimization steps. As shown in the Table, performance on HumanEval peaks at T=4 (54.88%), with T=3-5 forming an optimal range (53.83%-54.88%). Beyond T=5, performance gradually declines, suggesting over-adaptation. The consistent improvements across T=3-7 (all >52%) demonstrate our method's robustness. Note that we choose T=5 for all experiments in the paper, and do not finetune this value.
>
> ## Response to compare with test-time adaptation methods that introduce retrieval overhead.
> You also were wondering whether we compare to classical test-time adaptation, so "methods assume accessible training data during deployment and introduce significant retrieval overhead, limiting real-world practicality", as mentioned in our introduction. We do so; this is C3PO, which is a state-of-the-art retrieval-based test-time adaptation method. Our comparisons to C3PO can be found in Table 1 and Table 3.
>
>
> We hope our clarifications have addressed your concerns. Please don’t hesitate to reach out with any further questions. We would be happy to discuss further.

---

> > ### Author Response · Authors · 2025-11-26
> >
> > Thanks again for your feedback. We've added detailed clarifications on latency and included extra experiments for sensitivity analysis. Could you let us know if our responses address your concerns?

---

### Official Review · Reviewer_ccv4 · 2025-11-01

**Soundness:** 3
**Presentation:** 2
**Contribution:** 3
**Rating:** 4
**Confidence:** 4

**Summary:**

This paper proposes a test-time adaptation method for MoE models, which optimizes the router logits based on the cross-entropy loss of the input sequence and current generation itself. This allows the MoE routing to be adjusted based on the characteristics of the actual input data. Furthermore, by adapting continuously, the model can acquire more task-specific routing. The experimental results show modest performance gains across various tasks, with a notable improvement confirmed on HumanEval in particular, compared to the baseline.

**Strengths:**

This is a very simple and lightweight method, and it could be a promising option in scenarios where test-time adaptation proves effective. Since significant performance gains are achieved on highly specialized tasks like HumanEval, it suggests that this lightweight optimization alone may be sufficient even in situations requiring significant domain adaptation. Furthermore, because it is data-free and can be used even when reference data is not available, it offers many practical advantages.

**Weaknesses:**

- There are some points that seem unclear regarding the fairness of the experimental setup. Please refer to the Questions.
- It appears there are two errors in the optimization strategy. First, the paper states a policy of updating layers that performed 'high-confidence' routing. It seems this should be the opposite; layers with 'low' confidence should be updated. Second, regarding Equation (2), the paper claims, "higher confidence values indicate more decisive routing decisions." This formula, however, is equivalent to entropy, where a higher value actually represents greater uncertainty. It looks as though these two errors might be acting complementarily, coincidentally leading to the correct optimization strategy (i.e., updating the uncertain layers). However, this makes the paper's reasoning unreliable. The authors should probably re-verify that the results shown in the paper align with their intended methodology.

**Questions:**

- When applying this method, are the adaptation parameters (the $\delta$ vectors) re-initialized to zero for each test example? If it's not the case, it suggests that for all examples after the first one, a form of task-related leakage might be occurring. This could violate the fundamental assumption of evaluating samples independently and would also imply that performance becomes dependent on the evaluation order of the examples.
- If each example is completely independent, the proposed method should perform better in the latter half of long sequences. Are there any experimental results that have been verified for different lengths of examples?

---

> ### Author Response · Authors · 2025-11-20
>
> Thank you for your careful review. We're glad that you find it promising that light-weight optimization may be a workable strategy for test-time adaptation. We've addressed your questions below:
> ## Response to Optimization Strategy
>
> ### (1) Updating High-Confidence Layers
>
> You were wondering whether there is an error in the layer selection objective, and whether we actually update low-confidence/high-entropy routers. Based on your feedback we carefully re-checked our math, but we do indeed update high-confidence/low-entropy routers as we intended.
>
> Equation (2) measures confidence via the sum of log-probabilities (more precisely negative Burg's entropy, "`-log(p)`"), and not Shannon entropy "`-p log(p)`". Under our measure of confidence, higher values do correspond to lower entropy, e.g. if we illustrate in a 2D example:
>
>
> | Distribution | Our $C_i^{(n)}$ | Shannon Entropy $H$ |
> |--------------|-----------------|-------------|
> | (0.9, 0.1)   | 1.204          | 0.325       |
> | (0.5, 0.5)   | 0.693          | 0.693       |
>
>
> We intentionally update **high-confidence layers** rather than low-confidence ones. Our rationale was that these layers correspond to task-specific components that are actively selected during routing. To reduce computational cost at test time, we focus our updates on these critical layers.
>
> However we see that this may be counterintuitive! As, such the original submission also tested the opposite design (see "reverse metric" in Table 2), which performs noticeably worse, confirming that high-confidence layers are the appropriate optimization target. This strategy aligns with findings in recent related work [1], which similarly identifies critical layers for effective expert re-mixing.
>
> [1] Li Z, Li Z, Zhou T. C3po: Critical-layer, core-expert, collaborative pathway optimization for test-time expert re-mixing. arXiv preprint arXiv:2504.07964, 2025.
>
>
> Therefore, to summarize, there are **no complementary errors** in our methodology. Both our optimization strategy (targeting high-confidence layers) and our confidence metric operate as intended and are validated through ablation studies. Still we're glad that you brought this up, and we have changed our wording slightly to point out the potential for confusion of confidence as sum-of-log-probs and Shannon entropy.
>
> ## Response to  experimental setup
>
> You also wondered whether we introduce task leakage via the $\delta$ parameter. This is not the case.
> The $\delta$ parameters are re-initialized to zero for each test example and completely independent from each other, ensuring fair and independent evaluation across all samples. That it is possible at all to find test-time adaptations like this on a per-sequence level with no external data is a key finding of our work that we find particularly promising.
>
>
> ### Response to experiments on different lengths of examples
>
> To provide additional insights into how our adaptation scales with the length of the generated sequence, we have now conducted additional analysis on HumanEval by stratifying samples based on their sequence lengths and computing accuracy for each bucket.
>
>
> | Length Range | Ours | Baseline | Gain |
> |--------------|------|----------|------|
> | < 256 | 100.0% | 100.0% | +0.0% |
> | 256-384 | 73.5% | 70.6% | +2.9% |
> | 384-512 | 61.8% | 58.2% | +3.6% |
> | 512-640 | 46.3% | 43.9% | +2.4% |
> | > 640 | 23.3% | 10.0% | **+13.3%** |
>
> Interestingly, our method shows increasing gains on longer sequences, with the largest improvement (+13.3%) on samples exceeding 640 tokens. This validates that test-time routing optimization becomes more beneficial as context length grows, supporting our iterative refinement strategy.
>
>
> Thank you for your detailed comments. We think we were able to address your two key concerns, confirming that there is no error in the confidence computation, and that the delta parameters are indeed not shared between data points. Please don’t hesitate to let us know if you have any further questions. We would be happy to have more discussion.

---

> > ### Author Response · Authors · 2025-11-26
> >
> > Thanks again for your feedback. We made a detailed clarification to address two key concerns. We would greatly appreciate it if you could let us know whether our responses address your concerns or if any further clarification is needed.

---

### Meta-Review · Area_Chair_PweA · 2026-01-07

**Summary:**

1. Reviewer ccv4 and Reviewer ufCS question the confidence-based selection rule of the proposed method. In particular, Reviewer ccv4 finds two counter-intuitive parts (the reviewer called them errors) in the proposed optimization strategy: 1) the updating policy perform updates on the high-confidence layers instead of the low confidence layers, 2) the paper claims higher confidence values indicate more decisive routing decisions, but the formula used in this paper to indicate a high confidence value is similar to entropy, where a higher value represents a larger uncertainty. Reviewer ufCS also questions the underlying assumption that higher-confidence layers are more important for adaptation and hopes to see more empirical evidence for this assumption.


The rebuttal clarifies that they didn't use Shannon entropy, but uses Burg's entropy to measure the confidence, and states that higher Burg entropy will correspond to lower Shannon entropy, which is not correct and one can easily construct a counter example. For example, consider $p=(0.8, 0.1, 0.1)$ and $q=(0.6, 0.399, 0.001)$. The distribution $q$ enjoys both higher Shannon entropy and higher Burg entropy.

Based on the reviewer ccv4's comment and the rebuttal, I strongly suggest the authors re-check their updating rule.

2. Reviewers 94hV and Vpis question the additional computational overhead of the proposed method. For example, Reviewer 94hV points out that the proposed method nearly doubles the inference time compared to baselines on HumanEval. The rebuttal highlights that the proposed approach provides the optimal performance-efficiency tradeoff by taking advantage of the additional computation. However, there is no evidence for the optimality of the trade-off achieved by the proposed method, and even no definition of the optimal trade-off claimed in the rebuttal.

3. Both Reviewer ufCS and Reviewer Vpis raise questions on the difference between the optimization objective used for the updating rules and the actual goal. The objective is based on already-generated tokens, but the goal is to improve routing for the subsequent reasoning steps. Reviewer ufCS hopes to see some theoretical justification to support this choice of updating rule.
This is not addressed successfully in the rebuttal, as Reviewer ufCS pointed out in the discussion.

**Reviewer Concerns:**

Concerns addressed by the rebuttal:
- Reviewer ccv4’s concern on task leakage and evaluation independence.

 The rebuttal clarifies that the routing adaptation parameters are re-initialized to zero for each test example, ensuring independence across evaluation samples.

- Reviewer 94hV’s concern about the sensitivity

 The rebuttal adds an ablation study over the number of optimization steps $T$ to show a reasonable operating range and performance robustness.

Concerns unaddressed by the rebuttal:

- Reviewer ccv4 and Reviewer ufCS's concerns on updating the higher confidence layers and choice of the confidence measure.

The rebuttal clarifies that they didn't use Shannon entropy, but uses Burg's entropy to measure the confidence, and states that higher Burg entropy will correspond to lower Shannon entropy, which is not correct and one can easily construct a counter example. For example, consider $p=(0.8, 0.1, 0.1)$ and $q=(0.6, 0.399, 0.001)$. The distribution $q$ enjoys both higher Shannon entropy and higher Burg entropy.


- Reviewer ufCS and Reviewer Vpis's concern on the difference between the optimization objective used for the updating rules and the actual goal and the lack of theoretical justification for the proposed optimization objective.

The rebuttal clarifies that this is not a theoretical paper and only an empirical paper, but the rebuttal still fails to provide convincing evidence to support the choice of the proposed optimization objective, theoretical or empirical.

- Reviewers 94hV and Vpis's concerns on the additional computational overhead. The rebuttal claims that the proposed method achieves an optimal performance-efficiency tradeoff, without providing evidence for such optimality.

**Reviewer Scores:**

- Reviewer ccv4 (score 4): I don't think the reviewer will increase their score.

- Reviewer 94hV (score 2): I think the reviewer may keep this score or increase it to at most 4.

- Reviewer Vpis (score 6): I don't think the reviewer will change this score.

- Reviewer ufCS (score 4): I don't think the reviewer will change this score. The reviewer mentioned in the discussion phase that the rebuttal didn't address their central concern.

---

### Decision · Program_Chairs · 2026-01-26

Reject